# [Re] Parameterized Explainer for Graph Neural Network

## 1    Reproducibility Summary

### 2    Scope of Reproducibility

3   In this work we perform a replication study of the paper *Parameterized Explainer for Graph Neural Network*. The
4   replication experiment focuses on three main claims: (1) Is it possible to reimplement the proposed method in a different
5   framework? (2) Do the main claims with respect to the GNNExplainer hold? (3) Is the used evaluation method a valid
6   method for explaining the classification decision by Graph Neural Networks?

### 7    Methodology

8   The authors' TensorFlow code was largely used as starting point for our reimplementation in PyTorch. However, large
9   parts of the evaluation setup were missing and differences were found between the listed configurations in the paper and
10   the code. As a result, our codebase contains a large portion of novel code and introduces a different method for tracking
11   experimental configurations. Using the new codebase all experiments are replicated. In addition to this, a short ablation
12   study is performed.

### 13    Results

14   Due to numerous inconsistencies between code and paper, it is not possible to replicate the original results using the
15   paper alone. With help of the original codebase, a number of the original results can be retrieved. The main comparison
16   claim of the paper, to improve over the preceding GNNExplainer, does hold. However, after performing the replication
17   experiments, some questions regarding the validity of the used evaluation setup in the original paper remain.

### 18    What was easy

19   The method proposed by the authors for explaining the Graph Neural Networks is easy to comprehend and intuitive.
20   Re-implementation of the method is straightforward using a modern deep learning framework. The datasets used for
21   the experimental setup were all provided together with their codebase.

### 22    What was difficult

23   The main difficulty arose from the difference between the experimental configurations discussed in the paper and
24   implemented in the code. There were a number of small inconsistencies (eg. incorrect hyperparameter settings), but
25   also some major ones (eg. using batch-normalization in training mode during evaluation). This issue was worsened by
26   the fractured reporting of configurations in the code.

### 27    Communication with original authors

28   Contact was made with the authors on two occasions. During the first exchange the authors confirmed a number of
29   clarifying questions and confirmed that the configurations as presented in the codebase were to be used instead of
30   those provided in the paper. In the second exchange our reservations concerning the used experimental evaluation were
31   conveyed to the authors. The authors did not share our concerns.

## 1 Introduction

Graph Neural Networks (GNNs) emerged as state-of-the-art models in machine learning, capturing both graph structure and node features through recursively incorporating a graph's previous node information. GNNs are able to deliver state-of-the-art performances in matters such as graph/node classification and link prediction.

As for most Neural Networks, the 'reasoning' towards classification inside GNNs is not intuitive to humans. The authors of the paper *GNNExplainer: Generating Explanations for Graph Neural Networks* [9] address this problem and try to solve it by introducing GNNEXPLAINER; an optimization task that maximizes the mutual information between a GNN's prediction and a distribution of possible sub-graph structures. The GNNExplainer's algorithm can identify the sub-graph and node structure responsible for a given classification. Based on the work done in [9], Luo, D. et al. claim to have further developed GNNExplainer in their paper *Parameterized Explainer for Graph Neural Network* [5]. The paper introduces PGEXPLAINER; a general parameterized explainer that applies to any GNN based models in both transductive and inductive settings.

The authors of the paper first formulate the learning objective of PGExplainer. Using the same datasets as [9], they claim to outperform GNNExplainer up to $24.7\%$ in Area under the ROC Curve (AUC) [4] score. Furthermore the authors state that PGExplainer can speed up computations up to 108 times faster than GNNExplainer. These and further claims made in the PGExplainer paper will be evaluated in this report by replicating and extending the performed evaluation in a replication study.

**Scope of reproducibility** The focus of our reproducibility study is on the experimental comparison between the PGExplainer and the preceding GNNExplainer. The authors of the original PGExplainer paper include a number of other benchmarks in their evaluation, but focus their comparison primarily on the GNNExplainer. For this reason it makes sense for us to do the same.

In contrast to the original paper, we will base our entire comparison on reimplementations of both methods. In the original paper, the authors partly copy the results from the GNNExplainer and partly use their own re-implementation to obtain the GNNExplainer scores. In communication the authors stated that the decision to partly copy the results was made due to lackluster results in their own re-implementation. As the quality of an explanation is highly dependent on the model it aims to explain, we believe that it would be beneficial to re-implement both methods in the same framework and perform their evaluation on equal footing. We will use PyTorch as the framework for doing so.

For the reimplementation of the PGExplainer the authors' own TensorFlow-based codebase provided in their paper will be used as the main starting-point. However, during inspection of the codebase, we found that there are a number of significant differences between the configurations used for both the trained models that we wish to explain and the PGExplainer itself between what is described in the paper and what is actually implemented in the code. After discussing with the authors, the conclusion was reached that the configurations used in the code should serve as the starting point for the replication. Part of our reproduction experiment will focus on validating if these are indeed the correct configurations. In short, our replication experiment aims to validate the following aspects of the original paper.

(a) Given the original codebase and configuration files provided therein, is it possible to reimplement the PGExplainer method using a different framework? And if so, are the provided configurations sufficient to obtain the presented quantitative, qualitative and efficiency results.

(b) The authors claim that their PGExplainer greatly improves over the previously proposed GNNExplainer. We aim to validate that this claim holds with both methods evaluated using the same framework and evaluation.

(c) Evaluation of explanation methods is notoriously hard. We wish to validate if the evaluation method used in the original paper is a sound approach for doing so.

The remainder of this work will be structured as follows. In the next section we will provide the needed background on the PGExplainer. Following this, we will provide a short overview of the codebase accompanying this reproduction. In section 4, we will discuss the original experimental setup in depth and highlight some key components not discussed in the original paper. Section 5 will present the replicated results and compare them to the original paper. Based on the highlighted components in section 4 and some results presented in section 5, section 6 will raise some question regarding the evaluation setup used. In the last section, we will summarize our replication.

## 2 PGExplainer

The authors start by dividing an input graph $G_o$ in two subgraphs, such that $G_o = G_s + \Delta G$. $G_s$ represents the *explanatory graph* that makes important contributions towards the graph classification, while $\Delta G$ represents the

remainder of the initial graph. The main task therefore is to find the optimal subgraph $G_s$. This is achieved through Mutual Information ($MI$) maximization:

$$\max_{G_s} \text{MI}\left(Y_o, G_s\right) = H\left(Y_o\right) - H\left(Y_o \mid G = G_s\right), \tag{1}$$

Which uses the GNN's classification prediction $Y_o$ and its input $G_o$. The MI maximization is done by deducting the conditional entropy from the marginal entropy. Which is equivalent to minimizing the conditional entropy.

To avoid having an exploding exponential amount of candidates, the authors assume the explanatory graphs used are Gilbert random graphs [3], where selections of edges from the original input graph $G_o$ are conditionally independent to each other. Using relaxation, the learning objective is rewritten as

$$\min_{G_s} \mathbb{E}_{G_s}\left[H\left(Y_o \mid G = G_s\right)\right] \approx \min_{\Theta} \mathbb{E}_{G_s \sim q(\Theta)}\left[H\left(Y_o \mid G = G_s\right)\right], \tag{2}$$

where $q(\Theta)$ is the distribution of the parameterized explanatory graph. Each graph edge obtains a continuous variable in range $(0, 1)$.

A random graph $\hat{G}_s$ is sampled from edge distributions and fed to the trained GNN model obtaining prediction $\hat{Y}_s$. Following [9], the authors modify the conditional entropy with cross-entropy $H(Y_o, \hat{Y}_s)$, where $\hat{Y}_s$ is the prediction of the GNN model with $\hat{G}_s$ as input. Using Monte Carlo approximation, the learning objective becomes

$$\min_{\Omega} -\frac{1}{K} \sum_{k=1}^{K} \sum_{c=1}^{C} P_\Phi\left(Y = c \mid G = G_o\right) \log P_\Phi\left(Y = c \mid G = \hat{G}_s^{(k)}\right), \tag{3}$$

with $\Phi$ as the parameters in the trained GNN, $K$ as the number of sampled graphs, $C$ as the number of labels and $\hat{G}_s^{(k)}$ the $k$-th sampled graph, parameterized by $\Omega$.

Furthermore, PGExplainer is used to collectively provide explainations for multiple instances $\mathcal{I}$. The authors present the learning objective of this set of instances as follows.

$$\min_{\Psi} -\sum_{i \in \mathcal{I}} \sum_{k=1}^{K} \sum_{c=1}^{C} P_\Phi\left(Y = c \mid G = G_o^{(i)}\right) \log P_\Phi\left(Y = c \mid G = \hat{G}_s^{(i,k)}\right) \tag{4}$$

Here $\Psi$ are parameters in the explanation network, $G^{(i)}$ the input graph and $\hat{G}^{(i,k)}$ the $k$-th sampled graph for the $i$-th instance. Using the above, the authors consider two explainer instances; one for node classification and one for graph classification. Both cases use a MLP parameterized by $\Psi$.

# 3 Reimplementation of code

This section shortly summarizes the main structure of the code accompanying this reproducibility check and provides the information needed to reproduce the experiments presented. Our reimplementation of the PGExplainer is based on the PyTorch [6] framework. More specifically, it uses the third party extension of PyTorch for Graph Neural Networks called PyTorch-Geometric [2].

The codebase is structured for the two main tasks performed in this paper; training the GNNs that will be explained by the PGExplainer and performing a replication of the original experiments. Additional scripts are included for performing the evaluations presented in the appendix. Each script is self-contained, handling things such as loading the dataset, loading the correct model and setting the hyperparameters. Each of these things are predefined in `json` configuration files. For sections directly related to a part of the codebase we have added a link to the corresponding module on GitHub. (Links will be added after review)

## 3.1 Experiment configuration files [configs/selector.py]

The codebase contains a large number of predefined configuration files. These configuration files are the main working horse for making the experiments presented in this work reproducible. There are two different types of configurations, one for each of the two main tasks mentioned previously. Shared between tasks is the common occurrence of the dataset, model and seed used. If a task is to be performed a number of times to achieve an average, the seed is replaced with a list of seeds. A full description of the configuration file setup can be found in Appendix A.

As these configuration files provide a reliable source for all relevant information needed to perform our evaluation, we will—for the remainder of this paper—only disclose the information needed to comprehend the experiment. For

details irrelevant to understanding the results—e.g. the used learning rate and specific framework versions—we refer to the provided configuration and codebase[1]. We understand that this breaks the papers self-containment. However, we believe that regarding the balance between page restrictions and replicability completeness, separating the concern of replicability from paper to codebase is the correct way to go. A single source of replicability information also prevents inconsistencies between the paper and the code base. As the paper under consideration will highlight, this is a concern.

# 4 Experiment Setup

In this section we will introduce the setup of the experimental evaluation performed by the authors of the PGExplainer. While replicating their evaluation, we found that a number of steps were making assumptions that were not well documented. This includes the samples used for calculating the AUC score. In this section we will spend time on these steps. Additionally, some minor mistakes made in the original evaluation were rectified during our reproduction. These changes will also be highlighted here.

The experimental setup used by the authors of the PGExplainer follows that of the GNNExplainer [9] with a number of extensions. To clarify, the authors' proposed method serves the purpose of explaining the classification decision of a GNN. Hence, the experiments used to evaluate the PGExplainer focus on the explanations provided by the PGExplainer for the underlying model. Specifically, the evaluation is repeated for six different datasets, and thus, for six different underlying models. The six datasets span two different classification tasks; node-classification and graph-classification.

## 4.1 Datasets                                                         [datasets/dataset_loaders.py]

The node classification task is performed using four synthetic datasets (a-d). All of which are first introduced in the GNNExplainer paper [9]. The graph classification task is performed using two datasets (e-f), one synthetic and one real.

A reoccurring concept in all synthetic datasets is the so called *motif*. Motifs are highly structured subgraphs—e.g. 9 nodes connected in a 2D grid. These subgraphs are then expanded by attaching them to a randomly generated graph of a different structural form—e.g. Barabasi-Albert (BA) graph [1] or trees. Motifs play a crucial role in determining ground-truth explanations for our evaluations, as we will see later.

(a) The BA-Shapes dataset consists of single base BA-graph with 300 nodes, 80 "house"-structured motifs—each attached to random BA nodes—and some extra randomly added edges. (b) BA-Community closely resembles BA-Shapes, connecting two BA-Shapes and utilizing a Gaussian distributions for each BA-Shape to sample node features. (c) Tree-Cycles adopts an 8-level balanced binary tree as the base graph with a set of 80 six-node cycle motifs attached to randomly selected nodes. (d) The Tree-Grids dataset is similar to Tree-Cycles, replacing cycle motifs with $3 \times 3$ grid motifs. (e) The authors constructed the BA-2motifs dataset consisting of 1000 BA graphs. Half of the graphs contain "house" motifs, the other half contain five-node cycle motifs attached to the BA graph. These two types of graphs serve as the two classes for the dataset. (f) The real-life Mutagenicity dataset copied from [9], consisting of 4337 molecule graphs. These should be classified as either mutagenic or nonmutagenic.

## 4.2 Model                                                                  [models/GNN_paper.py]

There are a number of large differences between the implementation of the models trained for each dataset and how they are described in the paper. These changes are different between the node and graph classification tasks.

**Node classification**   The authors describe the model for node classification to be three consecutive Graph Convolution layers feeding directly into the fully connected classification. The model in the codebase however first concatenates the three intermediate outputs of the Graph Convolution layers before using this enlarged embedding as the input for the fully connected classification layer. The coded version of the models is similar to what is used for evaluation in the GNNExplainer paper [9]. To keep the evaluation consistent, we will therefore use the coded model version instead of the one described in the paper for our evaluation. Moreover, we were not able to get the model described in the paper to train to the same accuracy using the provided hyperparameters.

In addition to the architecture change, we found the node classification models to use an undocumented batch normalization layer after the first and second Graph Convolution layer. Unfortunately, the original codebase contained an error that resulted in these batch-normalization layers being kept in training mode during evaluation. This observation was confirmed by the authors in communication and has since been resolved. In the same communication the authors expressed that to be able to reproduce their results, the batch normalization layers will have to be kept in training mode.

---

[1]https://www.github.com/code_will_be_added_after_review

167 We believe that this will compromise the usability of our reproducibility experiment and therefore decided to remove
168 the batch normalization layers all together. For completeness full replication of the authors evaluation with a model
169 containing batch normalization is included in Appendix B.

**Graph classification**    The graph classification models are more in line with the models described in the paper than
171 the node classification models. The difference is the use of both max and mean pooling over the output of the final
172 Graph Convolution layer. These two pooling types are concatenated to form inputs for fully connected layers.

### 4.3   Evaluation metrics                                                       `[tasks/replication.py]`

174 For each dataset, the explanations are evaluated using three broad categories; quantitative, qualitative and efficiency.

#### 4.3.1   Quantitative evaluation                                        `[evaluation/AUCEvaluation.py]`

176 For each dataset the explanations provided by the PGExplainer are compared to ground-truth explanations. These
177 ground-truths describe for each sample which edges should or should not be included in the explanation. Using this
178 methodology, the quantitative evaluation can be performed similar to a binary classification task. For this reason, the
179 authors present the quantitative score using the AUC scoring metric.

**Ground Truth**    For node classification the ground-truth explanation is determined globally—i.e. for all node samples
181 the edges have the same ground-truth explanation label. Specifically, for each edge it is determined if the two nodes it
182 connects are part of a motif. When this is the case, the edge is labelled as positive for the ground-truth explanation.
183 Otherwise, the edge is labelled as negative for the ground-truth explanation. For graph classifications this is dependent
184 on the dataset used and how the ground-truth explanations are generated. For the BA-2motif dataset, being synthetic,
185 this is done the same way as for the node datasets. The only difference being that the process is repeated for every graph
186 in the dataset. As there are no motifs defined for the Mutagenicity dataset, the ground-truth labels can not be defined
187 based on them. Instead, for this dataset edge labels are used, as provided by the original dataset repository[2].

**AUC score**    With the explanation mask provided by the PGExplainer and the ground-truths defined as above, the AUC
189 score can be computed. However, there are a few important notes to consider when computing the AUC score. First, for
190 the node classification datasets, the explanation mask is only determined for a 3-hop graph around each node. This is
191 done because the GCN model only contains three layers. Second, only the nodes that are part of a motif are used in the
192 AUC computation. This is because there is no real definition of ground-truth for the nodes outside the motifs. This
193 evaluation design choice is further discussed in Sec. 6. Third, for the BA-2Motif dataset only a subset of the graphs is
194 used to determine the AUC score, this is done to reduce computation time. Lastly, for the Mutagenicity dataset only the
195 mutagenic graphs have a valid ground-truth interpretation. Hence, the AUC is determine using only these graphs. Of
196 these four considerations, only the last is mentioned in the original paper.

**Comparison**    The authors compare their method against four baselines; a gradient-based model (GRAD) [9], a graph
198 attention network (ATT) [8] and Gradient [7]. With the exception of the scores presented for the graph-classification
199 datasets, the scores presented are reused from the PGExplainer paper (see Table 4). In communication with the authors,
200 it was mentioned that the reimplementation of these explainers by the authors had resulted in lackluster results. For this
201 reason the decision was made to use the original scores by the original authors.

202 For our replication of the evaluation we focus our comparison on the GNNExplainer. This method is the most similar
203 and was a major inspiration for the PGExplainer. In contrast the the original evaluation, we do perform the comparison
204 using our own re-implementation of the GNNExplainer. Our re-implementation of this method is largely inspired by
205 the implementation in the PyTorch Geometric library. The main difference is that our re-implementation is adapted to
206 also work with graph-classification datasets. This is not possible with the plain PyTorch Geometric implementation.

#### 4.3.2   Qualitative evaluation                                               `[utils/plotting.py]`

208 In order to obtain a visualisation of the chosen sub-graph the system takes as input the ground truth labels and the
209 mask provided by the Explainer. Given the mask, two thresholds are calculated, one for importance to the explanation
210 and one to determine which other elements to plot for the sub-graph. Then, using these thresholds all nodes that
211 have an interesting enough weight are selected. Following this, only nodes that are in a direct sub-graph together the
212 node-to-be-explained are selected. When drawing the explanation for the graph classification this sub-graph is selected
213 using the top-$k$ edges. The original evaluation sets $k$ to be the number of edges in the defining motif for the synthetic

---

[2]https://ls11-www.cs.tu-dortmund.de/staff/morris/graphkerneldatasets

| Accuracy | Node Classification | | | | Graph Classification | |
|---|---|---|---|---|---|---|
| | BA-Shapes | BA-Community | Tree-Cycles | Tree-Grid | BA-2motifs | Mutagenicity |
| Training | 0.97 | 0.90 | 0.94 | 0.96 | 1.00 | 0.82 |
| Validation | 1.00 | 0.75 | 0.98 | 0.99 | 1.00 | 0.82 |
| Testing | 1.00 | 0.72 | 0.94 | 0.99 | 0.99 | 0.81 |

Table 1: Accuracies of the trained model without batch-normalization. The accuracies are obtained using early stopping.

datasets. These edges are plotted with a colour coding in accordance to their weight, where darker edges have higher weights in the mask than the lighter edges. Finally, the nodes that are connected to the previously plotted edges are plotted and colour coded by their ground-truth label.

### 4.3.3  Efficiency evaluation [evaluation/EfficiencyEvaluation.py]

In the paper, the authors only compare the efficiency of their PGExplainer to the GNNExplainer. Unfortunately, we were unable to extract the exact method for doing so from both the paper and the provided codebase. Our implementation is therefore mainly our own design.

We compute the inference time as the average over ten runs. During each run we measure the times it takes to explain all samples that are also used for the quantitative evaluation. This time is divided by the number of samples explained to get the final inference time per sample in milliseconds. Note that, similar to the paper, for the evaluation of the PGExplainer only the time to explain each sample is considered. On the other hand, for the GNNExplainer the time required to train the explainer is also taken into account because it has to be retrained for each sample.

## 5  Results

### 5.1  Model training [experiment_models_training.ipynb]

In Tab. 1 the final accuracies for all 6 trained models are provided. Note that these are the accuracies of the models that will be explained by the two explainers, not the explanation accuracy of the explainers themselves. For most of the models, using the configurations found in the code, we achieve results comparable to the results presented in the paper. The two exceptions being the BA-Community and the Mutagenicity models. Both of these score lower then their original counterpart.

Logically this difference could be contributed to the difference in the use of batch normalization. Where the original model in the PGExplainer paper did use batch normalization where we do not. However, as the results presented in Tab. 5 show, replication with the original batch normalization yields the same reduced accuracies. We hypothesise that therefore the difference might be the result of an undocumented use of weight regularization. We observed that in the original training script the configuration exist to use L2-weight regularization, but it is not used.

### 5.1.1  Replicability study [experiment_replication.ipynb]

**Quantitative** Quantitatively there is a large difference in the reported AUC scores and what we were able to achieve using the specified configurations for the PGExplainer. Only for BA-Shapes a AUC equal or higher then the presented AUC score was observed. However, BA-Shapes did require some minor modifications to the configurations to get it to work. With the temperature parameter set as originally presented in the code, the evaluation crashed. Only when the temperature was changed to the configuration as presented in the paper we were able to run the evaluation. Similarly, with the configuration as described in the code, the PGExplainer produces the opposite of the expected result for the BA-2Motifs dataset. This is reflected both quantitatively and qualitatively. However, it should be noted that the same drop in AUC score between our implementation and the one originally reported score can also be seen for the GNNExplainer. Due to this, the reported improvement of the PGExplainer over the GNNExplainer remains valid.

We believe that the difference seen between the AUC scores originally reported for the two explainers and what we observed during our reproduction might be the result of the undocumented effect of the entropy/size regularizations and used temperature. Based on empirical observations we found that the final AUC score is highly dependent on these three hyperparameters. A small follow-up ablation study presented in Tab. 5.1.1 confirms this.

**Qualitative** The replicated qualitative evaluation is very similar to the original results. PGExplainer is very capable of finding the motifs in the graphs and highlighting their edges. The same holds for the GNNExplainer.

| | Node Classification | | | | Graph Classification | |
|---|---|---|---|---|---|---|
| | BA-Shapes | BA-Community | Tree-Cycles | Tree-Grid | BA-2motifs | Mutagenicity |
| **Visualization (qualitative)** | | | | | | |
| PGExplainer | | | | | | |
| GNNExplainer | | | | | | |
| **Explanation AUC (quantitative** | | | | | | |
| Original | $0.963 \pm 0.011$ | $0.945 \pm 0.019$ | $0.987 \pm 0.007$ | $0.907 \pm 0.014$ | $0.926 \pm 0.021$ | $0.873 \pm 0.013$ |
| PGExplainer | $0.999 \pm 0.000$ | $0.825 \pm 0.040$ | $0.760 \pm 0.014$ | $0.679 \pm 0.008$ | $0.133 \pm 0.046$ | $0.843 \pm 0.084$ |
| GNNExplainer | $0.742 \pm 0.006$ | $0.708 \pm 0.004$ | $0.540 \pm 0.017$ | $0.714 \pm 0.002$ | $0.499 \pm 0.004$ | $0.587 \pm 0.002$ |
| Improvement | 34.6% | 16.5% | 40.7% | -4.9% | -375.2% | 43.6% |
| **Inference Time (ms) (efficiency)** | | | | | | |
| PGExplainer | 3.58 | 5.23 | 0.45 | 0.54 | 0.33 | 2.05 |
| GNNExplainer | 58.80 | 91.81 | 52.81 | 65.54 | 5.21 | 12.32 |
| Speedup | 16x | 17x | 117x | 121x | 16x | 6x |

Table 2: Replicated experimental results from the quantitative, qualitative and efficiency study. The original scores are copied from the paper directly. As the authors of the PGExplainer paper did not report the seeds used for the 10 validation results, we were unable to replicate these results using the authors own codebase. For the qualitative visualization the samples are handpicked similar to the original paper. Node colors represent the node labels (if all colours are the same the nodes are unlabeled). Darkness of the edges signals importance for the final classification decision. In case of the node-classification datasets the bigger node is the one for which the classification is being explained. For the quantitative explanation the average AUC score for the PGExplainer and GNNExplainer and the standard deviation is given. The "original" row reports the PGExplainer AUC score from the original paper. The inference time reported represents the time needed to explain a single sample in milliseconds.

| **Reg.** | | | | | **Size** | | | |
|---|---|---|---|---|---|---|---|---|
| | | | 10 | 1 | 0.1 | 0.01 | 0.001 | 0.0001 |
| | 10 | | $0.761 \pm 0.014$ | $0.761 \pm 0.014$ | $0.762 \pm 0.014$ | $0.713 \pm 0.156$ | $0.628 \pm 0.221$ | $0.634 \pm 0.239$ |
| | 1 | | $0.761 \pm 0.014$ | $0.760 \pm 0.014$ | $0.760 \pm 0.015$ | $0.683 \pm 0.154$ | $0.700 \pm 0.247$ | $0.708 \pm 0.226$ |
| **Entropy** | 0.1 | | $0.761 \pm 0.014$ | $0.760 \pm 0.014$ | $0.758 \pm 0.015$ | $0.565 \pm 0.246$ | $0.747 \pm 0.209$ | $0.764 \pm 0.214$ |
| | 0.01 | | $0.761 \pm 0.014$ | $0.760 \pm 0.014$ | $0.758 \pm 0.015$ | $0.551 \pm 0.249$ | $0.748 \pm 0.216$ | $\mathbf{0.776 \pm 0.210}$ |
| | 0.001 | | $0.761 \pm 0.014$ | $0.760 \pm 0.014$ | $0.758 \pm 0.015$ | $0.547 \pm 0.253$ | $0.753 \pm 0.216$ | $0.763 \pm 0.211$ |

Table 3: Results of a small ablation study on the effect of the size and entropy regularization on the AUC score. The ablation study is performed using the Tree-Cycles dataset and follows the setup of the quantitative evaluation. It averages over 10 runs. The results show that the regularization has a large effect on both the quantitative quality of the explanations and their consistency. The best score is shown in bold.          `[experiment_ablation.ipynb]`

The main observed difference is the Mutagenicity dataset. In our replication, only two edges are darkened in contrast to the ten edges darkened in the original paper. However, this difference is created artificially by a difference in the $k$ value reported in the paper and used in the code. While this difference therefore does not tell us anything about the quality of the explanation, it does show the importance of the $k$ hyperparameter. This is further discussed in the Sec. 6.

**Efficiency**   In terms of efficiency, the reimplemention results are consistent with the claims of the authors. The use of different frameworks between the original implementation and our reimplementation makes a direct comparison of the result is ill advised, but the speedup between the PGExplainer and the GNNExplainer is consistent.

# 6   Ground truth explanations for Graph Explanations

For the evaluation of the PGExplainer the authors made use of predefined ground-truth explanations. These explanations are made possible by the use of synthetic datasets, generated based on the notion of motifs. In this section we express some concerns with regards to the use of motifs for generating ground-truth explanations.

**No ground-truth outside motif**    In the case of the node classification datasets the definition of the ground-truth explanation is only valid for a small number of nodes within a graph; those within a motif. In essence, for nodes outside the motifs, the ground-truth explanation is an empty graph—i.e. all surrounding edges have to be excluded from the explanation to achieve the maximum score. The same is true for non-mutagenic graphs in the Mutagenicity dataset. This is incompatible with the PGExplainers approach to determine its explanation. An empty graph can never produce the same explanation as the original graph, hence it will never be the explanation provided by the PGExplainer.

The authors overcome this issue by excluding all nodes outside the motifs from their quantitative evaluation. However, this reduces the explanation task of the node classification datasets to a much simpler problem. For nodes outside the motif, the explanation has to be based on the absence instead of presence of edges. Solving these issues satisfactorily would require a new definition for the ground-truths for graph datasets. For example, in the case of the tree-cycle dataset, one could define the ground-truth of a node outside a motif to be the entire 7-hop subgraph as this would be the minimal number of steps to take before one can conclude that no cycles have been formed. We, however, believe this to be outside the scope of this replication.

**Qualitative evaluation dependent on knowing size of motif**    The PGExplainer gives as output a mask that describes for each edge in the graph the probability of it being important for the models classification decision. To turn this into a visualizable explanations the top-$k$ edges are selected from each mask, i.e only the $k$ edges that have the highest influence on the models classification decision are considered part of the explanation. As a result, $k$ is a crucial hyperparameter for obtaining a visual explanation. If $k$ is set too high, the explanation could contain edges that actually only contribute to the final decision marginally. If $k$ is set too low, the explanation could be missing important parts of the graph. This difference in visual explanation quality was also empirically observed in the difference between the original and our explanations for the Mutagenicity dataset.

As mentioned in the experimental setup of the qualitative evaluation, the authors, and preceding works, set the value of $k$ in the evaluation based on the amount of edges in the defining motif. However, this is not a possibility outside of the synthetic evaluation datasets. Hence, for real world applicability of the proposed explanation method a different approach has to be found to find $k$. For this reason, we believe that evaluating the quality of the explanations based using $k$ preset to the number of edges in the synthetic dataset is an aspect to reconsider.

In essence, both the $k$-parameter and the earlier mentioned number of edges selected for the ground-truth can be considered as hyperparameters for the evaluation pipeline. By selecting a specific value for these parameters the evaluation can become biased towards assigning high credibility to explanations that have a specific characteristic. By performing an extensive search over these hyperparameters the results of the explanation evaluation can potentially be improved. In Sec. C of the appendix we present a short study on how these hyperparameters can influence the final results of the evaluation.

## 7   Conclusion

In this work, we have presented a replication of the paper *Parameterized Explainer for Graph Neural Network*. The replication experiments have lead us to a number of conclusions. First, based on the paper alone, it is difficult to replicate the presented results. The main contributing factor is the discrepancy between the provided details in the paper and those in the codebase. Based on communication with the authors, we conclude that the hyperparameter settings presented in the paper are oversimplified. For the method to work, more hyperparameter tuning is needed then the paper suggests. This is validated by our ablation study.

Second, even with the provided codebase, replication of the presented results is still arduous. With the configurations pulled from the codebase used in our re-evaluation, we still found lackluster results for a number of the datasets. We accredit this problem mainly to the structure of the codebase itself. The code is overly convoluted with the experiment configurations being overridden in numerous locations. Due to this, it is unclear if the configurations we found in the codebase are those that generated the results presented in the original paper.

Lastly, as discussed in Sec. 6, we are uncertain if the evaluation based on synthetic datasets as used in the evaluation is valid. However, we can not contribute this issue to only the authors' paper as it is also used in other graph explanation papers, including the GNNExplainer. In addition to showing that these issues exists, our extended evaluation presented in appendix Sec C showed that it is not trivial to solve them based on the current definition of a ground-truth explanation for motif graphs. Rethinking the evaluation for Graph Neural Networks Explainers is therefore important future work.

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

## Appendices

## A    Original data and results PGExplainer

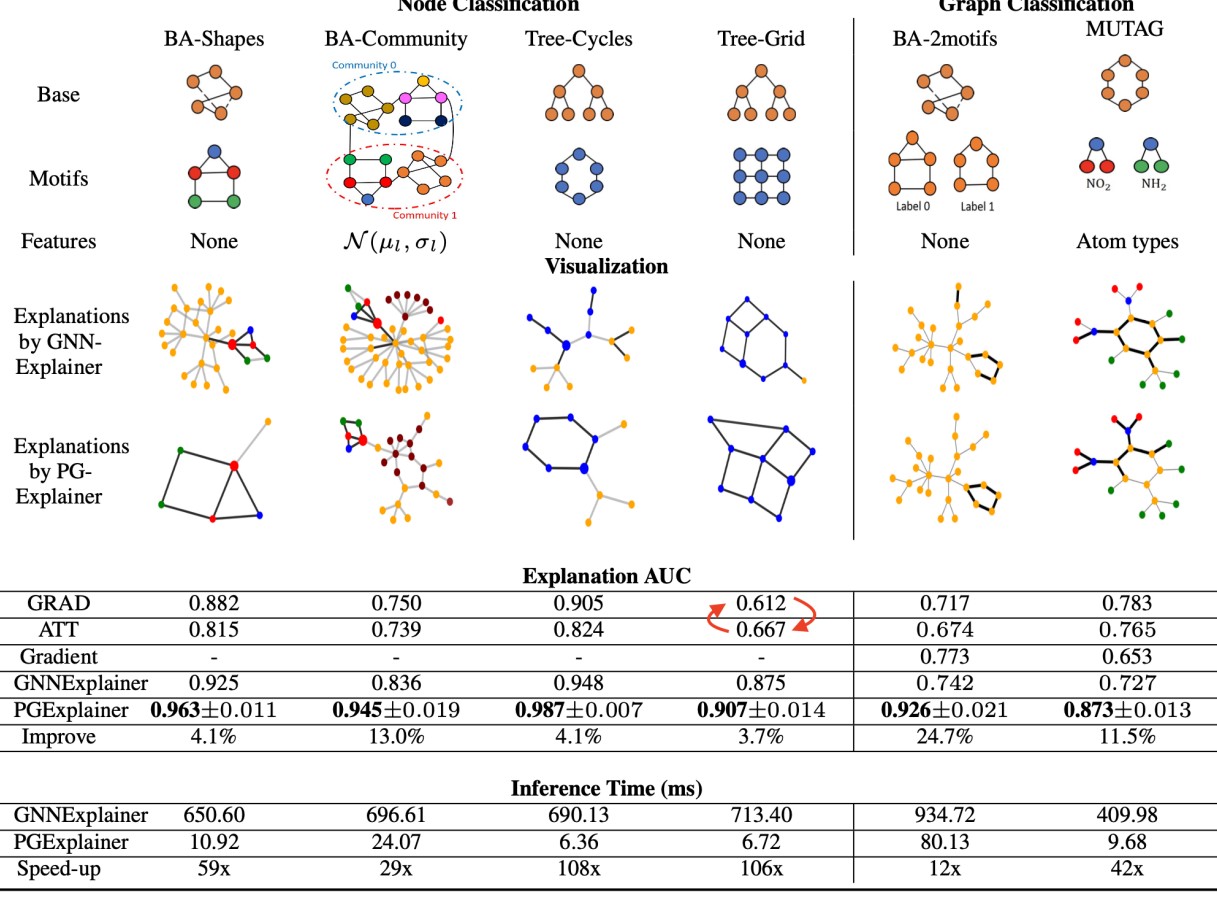

Table 4: Visual representations of the datasets, results and the performance evaluations [5]. **Note: The AUC scores for *GRAD* and *ATT* inside Tree-Grid are incorrectly copied by the authors and should be swapped (as indicated by the red arrows).**

## B    Direct replication of PGExplainer with BatchNorm activated model

Here we present a replication experiment similar to the one presented in our replication work. However, as discussed in in the main work, the models used in the original paper contained two batch-normalization layers. These layers were incorrectly kept in training mode during evaluation. In the replication results presented here, the same batch-normalization setup was used for the node-classification models.

### B.1    Results

**Model training**    Tab. 5 shows that the accuracies of the models trained using the batch normalization are very similar to those used in the main paper. The BA-Community dataset still shows the same issues with overfitting as are discussed in the main paper.

**Quantitative**    Quantitatively there is a significant difference between the explanations of the PGExplainer for models trained with or without batch normalization. However, the main conclusion based on these results remain the same. The

| | Node Classification | | | |
|---|---|---|---|---|
| Accuracy | BA-Shapes | BA-Community | Tree-Cycles | Tree-Grid |
| Training | 0.98 | 0.94 | 0.96 | 0.96 |
| Validation | 0.99 | 0.74 | 0.99 | 0.98 |
| Testing | 1.00 | 0.71 | 0.97 | 0.99 |

Table 5: Accuracies for models trained with batch-normalization. For evaluation of the validation and test dataset batch-normalization is kept in training mode. This is similar to the original paper.

| | Node Classification | | | |
|---|---|---|---|---|
| | BA-Shapes | BA-Community | Tree-Cycles | Tree-Grid |
| **Visualization** | | | | |
| Original | |  | | |
| No batch-norm | |  | | |
| With batch-norm | |  | | |
| **Explanation AUC** | | | | |
| Original | 0.963 ± 0.011 | 0.945 ± 0.019 | 0.987 ± 0.007 | 0.907 ± 0.014 |
| No batch-norm | 0.999 ± 0.000 | 0.825 ± 0.040 | 0.760 ± 0.014 | 0.679 ± 0.008 |
| With batch-norm | 0.977 ± 0.006 | 0.970 ± 0.006 | 0.534 ± 0.186 | 0.649 ± 0.045 |
| **Inference Time (ms)** | | | | |
| No batch-norm | 3.58 | 5.23 | 0.45 | 0.54 |
| With batch-norm | 3.56 | 5.29 | 0.40 | 0.47 |

Table 6: Replicated experimental results from the quantitative, qualitative and efficiency study. For the qualitative visualization the samples are handpicked similar to the original paper. Node colors represent the node labels (if all colours are the same the nodes are unlabeled). Darkness of the edges signals importance for the final classification decision. In case of the node-classification datasets, the bigger node is the one for which the classification is being explained. For the quantitative explanation the average AUC score and standard deviation is given. The "original" row reports the PGExplainer AUC score from the original paper. The inference time reported represents the time needed to explain a single sample in milliseconds.

replication experiments show that using the configuration provided in the codebase it is not possible to directly replicate the results presented in the paper.

**Qualitative** No consistent significant difference in the visualized explanations can be observed between the two explained models.

**Efficiency** The time required to explain the classification decision of a single node in the graph is consistent between the models trained with and without batch normalization.

# C Extended replication

In this extend replication we perform a simple experiment considering the issues raised in Sec. 3. Specifically, we redo the quantitative evaluation of the synthetic node-classification using all test nodes instead of only those located in a motif. The model used for the explanation and the definition of the ground truth remains the same.

| | Node Classification | | | |
|---|---|---|---|---|
| | BA-Shapes | BA-Community | Tree-Cycles | Tree-Grid |
| **Explanation AUC** | | | | |
| PGExplainer | 0.974 ± 0.005 | 0.576 ± 0.024 | 0.748 ± 0.014 | 0.790 ± 0.009 |
| GNNExplainer | 0.508 ± 0.008 | 0.555 ± 0.002 | 0.482 ± 0.014 | 0.608 ± 0.009 |

Table 7: Results of the extended replication study. For each explanation model both the AUC score over ten runs and the corresponding standard deviation is given.

## C.1 Results

Quantitatively the PGExplainer scores significantly worse in the extended replication than during the original replication (see Tab. 7). This is a direct result of performing the evaluation over the entire test set instead of only the nodes within a motif. The ground-truth for nodes outside the motif and the method used by both the GNNExplainer by PGExplainer are simply incompatible.

Nevertheless, the improvement claimed by the authors of the PGExplainer over the GNNExplainer is still visible. Considering all datasets, the PGExplainer consistently outperforms the GNNExplainer by a significant margin.

## D Configuration

Configuration files are used to provide a stable, flexible and reproducible way to run the experiments.

## D.1 Model configuration files

The first type of configuration is the model configuration as seen in Fig. 1, which contains (from top to bottom) the parameters required for training a GNN model. `dataset` assigns which dataset the model has to train on, `paper` defines which paper the model is build on (either PG, GNN or TAG), `lr` is the learning rate, `epochs` is the amount of epochs used for training and `clip_max` is the parameter to which determines at what point the gradient is clipped. Additionally the file includes `early_stopping` which defines the amount of epochs with no improvement that are required to enact early stopping of training, `seed` which defines the seed used for training and `eval_enabled` which determines whether the model uses it's eval mode.

```json
{
    {
    "model": {
        "dataset": "syn1",
        "paper": "GNN",
        "lr" : 0.001,
        "epochs" : 1000,
        "clip_max" : 2.0,
        "early_stopping": 100,
        "seed" : 42,
        "eval_enabled" : true
    }
}
```

Figure 1: Example of model config file in JSON format

## D.2 Explainer configuration files

The second type of configuration is the explainer configuration an example of which is shown in Fig. 2, these configuration files contain all parameters required to train an explainer and perform the the replication experiment using them. It includes the following parameters: `dataset` defines the dataset that the model that is to be explained is trained on. `model` is the type of model that has to be explainer, `explainer` is the implementation of the explainer (either PG or GNN). The configuration also contains the learning rate and number of training epochs (`lr` and `epochs` respectively). As well as `sample_bias` which determines the sample bias, the parameters `reg_size` and `reg_ent` that determine the size loss and entropy loss coefficients respectively, the temperatures, `seeds` the seeds used for training, `eval_enabled`

if the model uses evaluation mode and `thres_snip` and `thres_min` which define the thresholds for the interesting and sub-graph edges related to the drawing of the result explanations.

```json
{
    "explainer": {
        "dataset": "syn1",
        "model": "GNN",
        "explainer": "PG",
        "lr" : 0.003,
        "epochs" : 10,
        "sample_bias" : 0.0,
        "reg_size" : 0.05,
        "reg_ent" : 1.0,
        "temps" : [5.0, 2.0],
        "seeds" : [0, 1, 2, 3, 4, 5, 6, 7, 8, 9],
        "eval_enabled" : true,
        "thres_snip" : 12,
        "thres_min" : 100
    }
}
```

Figure 2: Example of explainer configuration file in JSON format

