# OpenReview forum: "[Re] Parameterized Explainer for Graph Neural Network"
_ML_Reproducibility_Challenge/2020 — RC2020_

### Official Review · AnonReviewer1 · 2021-03-01
**The report reveals a lot of dark spots of the original paper**

**Rating:** 7
**Confidence:** 4

**Review:**

As a first remark, the report seems to repeat content. This is probably not the author's fault, I noticed the same problem with other reports. That is probably a problem with the instruction template. This is something for the organizers to take into consideration.

General remarks
The report follows the prescribed format. The authors reproduced the experiments in pytorch. The original implementation was in tensorflow. The author found errors in the implementation and a lot of gaps in the parameter tuning. In some cases, they get results opposite from the claims of the paper. The biggest problem of the paper they evaluated was the fact that the text and code were not in agreement. In fact, the code was convoluted and it was difficult to actually get the knowledge out of it.
The report is well organized and there is a clear correspondence between the code and the issues the report addresses

Problems of the report
The report has a lot of references to nowhere, this is probably a problem with their latex referencing system. There are references to sections, tabs, and an appendix. While this is a technicality that can easily be resolved for the tables and sections, I couldn't locate the appendix. As mentioned in the comments the authors point that the pytorch function for the node convolution in pytorch is different from the one used in tensorflow. The original paper authors attribute the differences to that. In my opinion, this difference can not be responsible for the discrepancies.

**Familiar With The Original Paper:**

I have not read the original paper

**Reproducibility Summary:**

Report has summary

---

### Official Review · AnonReviewer2 · 2021-03-03

**Rating:** 7
**Confidence:** 4

**Review:**

The report is well written and has a concise explanation. Additionally, the authors provide a great summary at the beginning. I would like to know if the numbers from the TensorFlow implementation (authors' paper) match the results reported in the main paper. It would interesting to perform a hyperparameter search and provide code and docs so it would be reviewed before accepting the paper. After the authors of the report make some claims and report an improvement for their own implementation. I suggest accepting the report

**Familiar With The Original Paper:**

I have read the original paper

**Reproducibility Summary:**

Report has summary

---

### Public Comment · ~Dongsheng_Luo1 · 2021-02-18
**Usage of GCN layer**

Dear Authors,

Thanks a lot for your efforts conducting comprehensive experiments to compare PGExplainer and GNNExplainer and pointing out some issues in our paper and code.  I am glad to see that with your implementation, our PGExplainer still outperforms GNNExplainer.  Your extensive experiments definitely provide deep insights into our PGExplainer.

I have checked the attached code. It seems that you adopt GCNConv from torch_geometric, where each layer can be represented by
$f(H^{(l)},A)=\sigma(AH^{(l)}W^{(l)})$.

However, the GCN layer used in our PGExplainer and GNNExplainer is slightly different:
$f(H^{(l)},A)=\sigma(W^{(l)}AH^{(l)})$.
Please refer to
https://github.com/RexYing/gnn-model-explainer/blob/master/models.py#L70
and
https://github.com/flyingdoog/PGExplainer/blob/master/codes/layers.py#L37
Sorry that we didn't clearly state this detail in our paper.
As a result, the explained GNN models in this report are different from the ones in PGExplainer and GNNExplainer paper, which may explain different performances between our Tensorflow implementation and the Pytorch implementation.


Thanks again!

---

> ### Author Response · Authors · 2021-03-18
> **Impact of difference in GCN layer used should not be this large**
>
> Dear Dongsheng Luo,
>
> Thank you very much for your comment. We very much appreciate your involvement. This includes the clarifications you provided regarding the differences in the configurations while we were working on the report.
>
> We do indeed use torch_geometric which uses a slightly different GCN layer to construct the GNNs used for our evaluation. Hence, the models explained by the PGExplainer and the GNNExplainer are slightly different from those used in your own evaluation. This is the most likely explanation for the differences observed in accuracy training/validation/test accuracy for the models trained on the BA-Community and the Mutagenicity dataset.
>
> However, we are not convinced that this difference is the explaining factor for the difference observed between the results of the quantitative and qualitative analysis of the PGExplainer method itself. Ultimately, the difference is only in the underlying model that is being explained by the PGExplainer, not in the PGExplainer model used for generating the explanations. With the PGExplainer claimed to be a universal explainer for GNN models, we believe that the difference in the to be explained model should not have such a severe impact on the usefulness of the explanation.

---

### Decision · Program_Chairs · 2021-03-31

**Decision:**

Accept

**Comment:**

Selected for ReScience-C Journal Publication.

Good reviews, strong reproducibility report, provides code reimplementation from scratch which is a strong contribution. The authors have responded to reviewer comments and have updated their paper accordingly.